# DualTune-GhostDP: A Unified Framework for Synergistic Differentially Private Fine-Tuning of Large Language Models

## Abstract

With growing concerns about data privacy and confidentiality, there has been increased attention on privacy preserving integration in many applications, particularly data driven ones like Large Language Models (LLMs). LLMs are powerful in-context learners and are widely adopted in real world products. However, their dependence on sensitive private data in training and prompts exposes them to potential data leakage and privacy breaches. Differential Privacy (DP) delivers a rigorous, mathematically provable safeguard against these vulnerabilities; however, this assurance often comes with considerable reductions in model performance and increased computational cost. While prior work has highlighted the inherent trade-off between privacy and utility, our proposed method, DualTune-GhostDP, shows that strong privacy guarantees can be maintained under a controlled budget without sacrificing high model performance. Our method adopts a two-phase fine-tuning pipeline that integrates Ghost Clipping with an EdgeWorth (EW) Advanced Privacy Accountant, replacing conventional DP accounting mechanisms. Experimental results show that the principled integration of these components in DualTune-GhostDP consistently outperforms the individual benefits of each and both the single-phase Differentially Private Stochastic Gradient Descent (DP-SGD) baseline and a two-phase fine-tuning variant using standard clipping. Specifically, it achieves higher accuracy, faster convergence, and improved computational efficiency while maintaining DP guaranties. In addition, we assess robustness to Membership Inference Attacks (MIA), which aim to determine whether a particular sample was used during training. Our findings demonstrate that DualTune-GhostDP substantially mitigates membership leakage across all training stages, strengthening both privacy assurances and overall stability of the approach against such attack relative to existing baselines.

## 1 Introduction

With advancements in AI and NLP, LLMs including encoder-only, decoder-only and encoder-decoder models have demonstrated extraordinary capabilities across multiple domains, including language generation, sentiment analysis, question answering, and text classification (Das et al. 2025). Their ability to generate and understand human-like contextual language has fostered significant advancements across diverse sectors and transformed multiple industries (Trummer 2023). However, alongside these advancements, LLMs introduce significant privacy concerns (Gupta et al. 2024).

LLMs rely on vast amounts of data that often contain personal or sensitive information, increasing the risk of inadvertent memorization and data leakage. Recent studies show that private information can leak from LLMs, leading to growing interest in privacy-preserving techniques (Duan et al. 2024; Shi et al. 2021; Li et al. 2021a; Shi et al. 2022; Hong et al. 2023; Behnia et al. 2022). These risks are further exemplified by MIA, where adversaries exploit model outputs to determine whether a specific record was included in the training data. While MIA does not directly reconstruct sensitive records, it reveals weaknesses in how models handle training data and highlights that LLMs may overfit or retain information about individual samples, thereby undermining data confidentiality and user privacy (Carlini et al. 2022b). To mitigate such threats, various approaches have been proposed, with Differential Privacy (DP) emerging as a leading solution. However, as LLMs and attack strategies continue to evolve, ensuring strong privacy guarantees remains an ongoing

challenge. Existing DP-based methods for LLMs, reviewed in Section 3, often suffer from a privacy–utility trade-off that leaves room for further optimization.

Building on prior fine-tuning with DP work Li et al. (2021b); Yu et al. (2021), we propose *DualTune-GhostDP*, a unified framework that combines a two-phase fine-tuning strategy Shi et al. (2022), PromptDP-SGD soft-prompt optimization Duan et al. (2023), Ghost Clipping for memory-efficient DP-SGD training Li et al. (2021a), and the Edgeworth (EW) privacy accountant from EW-Tune Behnia et al. (2022). Unlike prior work, which studies these components independently, DualTune-GhostDP jointly integrates them into a single end-to-end framework, improving utility, convergence, memory efficiency, and privacy accounting while maintaining rigorous differential privacy guarantees. Our main contributions are summarized as follows.

- We propose **DualTune-GhostDP**, a unified framework for DP fine-tuning of prompt-based LLMs that integrates a two-phase training pipeline with Ghost Clipping and an EW privacy accountant.

- We demonstrate that separating task adaptation and private optimization reduces sensitivity and improves the privacy–utility trade-off compared to conventional one-phase DP-SGD training.

- We provide an extensive evaluation across multiple datasets, including utility, runtime, efficiency, gradient sensitivity, and MIA.

- We present ablation studies showing the individual contribution of the two-phase training strategy, Ghost Clipping, and the advanced privacy accountant.

The remainder of this paper is organized as follows. Section 2 presents essential background concepts, while Section 3 reviews related work. Section 4 describes the proposed approach, including its motivation, building blocks, and threat model. Section 5 presents experimental results and discussion, followed by an evaluation of robustness to MIA in Section 5.6. Section 6 concludes the paper and outlines future directions. The code for our experiments is publicly available at `https://github.com/AnonymousSatMLll/Differentially-Private-Prompting-in-Large-Language-Models.git`.

## 2 Background and Preliminaries

### 2.1 The Hype of Large Language Models

AI allows machines to perform tasks that traditionally require human intelligence. Within this scope, NLP focuses on understanding and generating human language, leading to the development of LLMs. LLMs leverage transformer-based deep learning architectures to process and generate text with remarkable fluency.

LLMs have gained attention in both academia and industry for their ability to solve diverse general-purpose tasks, rather than being limited to specific applications which made them highly versatile (Chang et al. 2024). They are trained on vast datasets across various domains and require significant computational resources (Myers et al. 2024). The different types of LLMs excel in tasks ranging from simple sentence classification, question-answering, and sentiment analysis to more complex applications such as sentence completion, conversational AI, and advanced text generation (Anisuzzaman et al. 2024).
A key aspect of using LLMs is fine-tuning, which involves taking a pre-trained model and retraining it on domain-specific data to improve efficiency and performance for specialized tasks. Different fine-tuning techniques have been developed to adapt to specific applications (Parthasarathy et al. 2024).

### 2.2 Differential Privacy (DP)

As data usage continues to grow, the protection of sensitive and private information requires a rigorous approach. DP, introduced in by Dwork (2006), provides a mathematical guarantee that ensures that an attacker, regardless of their computational power or access to data, cannot extract meaningful information about any individual.
DP guarantees that an individual's participation in a study or model has minimal impact on the outcome. In other words, any information inferred about a person comes solely from the model's output, not from

their direct involvement. Consequently, DP makes it difficult for adversaries to infer sensitive information. The core idea behind DP is the controlled addition of noise to model outputs or query responses. Given two neighboring datasets, $D$ and $D'$, which differ by only a single data point (through addition, removal, or substitution), DP guarantees that the statistical behavior of a mechanism responding to queries (i.e., the curator's method of answering) remains nearly identical while still providing useful responses. This in turn ensures that the answer does not reveal whether the mechanism was applied on $D$ or $D'$.

DP can be applied in two primary ways: Global DP (GDP) and Local DP (LDP), but our focus will be on GDP. **Global DP**: The noise is added centrally by a trusted server. **Local DP**: Users perturb their data locally before sending it to an untrusted server, ensuring privacy even when the server is not trusted.

**Definition 1** (($\epsilon, \delta$)-DP). *Given $\epsilon \geq 0$ and $\delta \geq 0$, a randomized algorithm $M$ is ($\epsilon, \delta$)-differentially private if, for all adjacent datasets $D, D'$ (differing by a single data sample) and all measurable subsets $S$ in the output space, the following condition holds:*

$$\Pr(M(D) \in S) \leq e^\epsilon \Pr(M(D') \in S) + \delta.$$

*$\epsilon$ is termed the privacy budget. For both $\epsilon$ and $\delta$, smaller values correspond to stronger privacy guarantees.*

### 2.3 Privacy Accountants and their Role in DP

DP mechanisms provide mathematical privacy guarantees, but they must be carefully tracked across repeated queries or training steps, as each application consumes part of the overall privacy budget. Without proper monitoring, cumulative privacy loss may exceed the allocated budget and breach the promised guarantees. To address this, *privacy accountants* were introduced to measure and track cumulative privacy loss, ensuring it remains within a predefined bound while enabling fine-grained control over different operations.

Several types of privacy accountants have been proposed for different mechanisms and noise distributions, including the Moments Accountant (Abadi et al. 2016), Rényi divergence-based accounting (RDP), which offers tighter bounds and flexible choice of divergence order $\alpha$ (Wang et al. 2019), Privacy Random Variable (PRV) accounting (Gopi et al. 2021), and specialized accountants for Gaussian (Koskela et al. 2022) and Laplace mechanisms (Dwork et al. 2006). Among these, RDP has become widely adopted in practice due to its strong privacy bounds and flexibility. More recently, the **Edgeworth Privacy Accountant (EW)** was proposed in (Wang et al. 2022), further improving over RDP by providing tighter finite-sample privacy estimates and better adaptability to practical DP training settings, as summarized in Table 6 in the Appendix A.1. The choice of privacy accountant ultimately depends on system requirements for accuracy, scalability, and implementation complexity.

## 3 Related Work

The success of LLMs in generating and processing information has impacted nearly every sector of our life with remarkable capabilities (Duan et al. 2023; Devlin et al. 2019; Achiam et al. 2023; Zhang et al. 2022). However, this advancement has also raised critical privacy concerns, posing risks to personally identifiable information and the potential leakage of sensitive data (Singh et al. 2024). In LLMs, the Prompting paradigm can be broadly studied under two paradigms: white-box and black-box settings. Our work operates in the white-box paradigm, where access to model gradients is available and fine tuning is duable, so this section reviews works primarily in this direction.

Several studies have focused on prompt fine-tuning in the white-box setting, where users have access to the model gradients. Li et al. (2021a) addressed the challenge of integrating DP-SGD into LLMs while maintaining both performance and computational efficiency. They investigated the use of large pretrained language models, such as RoBERTa, and emphasized the critical role of selecting appropriate hyperparameters for optimal performance. Contrary to non-private setups where smaller batch sizes and learning rates are preferable, their findings demonstrated the advantages of using larger values in the private fine-tuning setting. Additionally, to improve memory efficiency, they introduced ghost clipping to optimize memory usage when fine-tuning large transformers under DP-SGD. This technique extends the method in (Lee & Kifer 2020) while avoiding the instantiation of per-example gradients, even for individual linear layers.

In conventional DP-SGD, gradients are computed for each sample, individually clipped to a predefined bound $C$, and then averaged with added Gaussian noise. Ghost clipping replaces explicit per-sample gradient computation with an analytical approximation of each sample's gradient norm, referred to as the *ghost norm*. The ghost norm $g_i$ for a sample $i$ is estimated using the activations and back-propagated gradients as $\|g_i\|_2^2 = \sum_l \|J_l(x_i) \cdot \delta_l\|_2^2$. where $J_l(x_i)$ denotes the Jacobian of the $l$-th layer activations with respect to the input, and $\delta_l$ represents the back-propagated error. This estimate enables efficient clipping of gradients without instantiating them individually in memory, thereby substantially reducing computation time and GPU usage. After estimating the ghost norms, gradients are rescaled by $\min(1, \frac{C}{\|g_i\|_2})$ and aggregated across the batch before Gaussian noise is added, consistent with the standard DP-SGD update rule. Experimental results on sentence classification datasets in this work, confirmed the effectiveness of this approach reducing memory consumption by at least a factor of 22. The study demonstrated that leveraging large pretrained models, well-chosen hyperparameters, and direct application of DP optimization during fine-tuning leads to high-performing DP language models, even under a moderate privacy budget, achieving a strong balance between privacy and utility.

Another line of work introduced Just Fine-Tune Twice "JFT" (Shi et al. 2022), a framework that involves fine-tuning the model twice to safeguard against privacy leakage. The first fine-tuning phase is performed using redacted in-domain data, where sensitive information is concealed and no DP optimizer is applied. The second fine-tuning phase is conducted with the original data under a private mechanism. The proposed framework achieves Selective Differential Privacy (SDP), an extension of DP formalized by Shi et al. (2021), based on the understanding that sensitive information is typically sparse. SDP defines neighboring datasets to differ only in the sensitive part of a training example and as a result, SDP selectively hides the difference in the sensitive part only where they stated that it's particularly suitable for NLP tasks. In the first phase, redacted data is extracted from the original data using a secret detector. Based on the detector's performance through recall metric evaluation between the redacted data and the original, three methods are then employed to fine-tune the redacted data. If the detector successfully masks all sensitive information so recall score of 100, the model is fine-tuned directly with a public unnoised optimizer. If the detector is imperfect, an affordable subset is manually screened and fine-tuned using the public optimizer. In cases where the detector is imperfect and manual screening misses some sensitive data, light noise is added, and a private optimizer is used, where the missed sensitive information receives smaller epsilon values compared to the rest of the redacted data. After fine-tuning on the redacted data, the model is further fine-tuned on the original data using DP-SGD with a private optimizer. For their experiments, the authors used natural language understanding (NLU, on GLUE) and language generation datasets, demonstrating that JFT outperforms DP-SGD and Redacted-only models.

Building on DP fine-tuning, EW-Tune Accountant (Behnia et al. 2022) improved privacy accounting by leveraging Edgeworth approximations for finite-step training. This method tightens privacy guarantees and optimizes the noise multiplier, thereby reducing the utility loss typically incurred in DP-SGD. Experimental results on GLUE benchmarks demonstrated up to a 1.1% accuracy gain with reduced noise levels, showing its effectiveness for practical DP fine-tuning.

Some hybrid works also consider overlapping paradigms where gradient-based and gradient-free methods overlap (Duan et al. 2023). In this work, they proposed PromptDP-SGD, a gradient-based method that privately tunes soft prompt embeddings, achieving comparable performance to private fine-tuning but with reduced storage and training costs. In cases where gradients are unavailable, they extended the approach with PromptPATE, a black-box compatible method, though the white-box variant remains directly relevant to our setting.

While black-box approaches such as DP-OPT (Hong et al. 2023) and DP-GTR (Li et al. 2025) offer privacy-preserving solutions when gradients are inaccessible, they fall outside the scope of our contribution. Our focus remains on the white-box paradigm, where gradient access enables more direct integration of privacy-preserving optimizations.

Safeguarding LLMs is still an ongoing challenge, due to its devastating impacts. Existing white-box solutions address memory efficiency (ghost clipping), selective privacy (JFT), and tighter privacy accounting (EW-Tune). However, these contributions remain fragmented, with no unified approach combining their strengths.

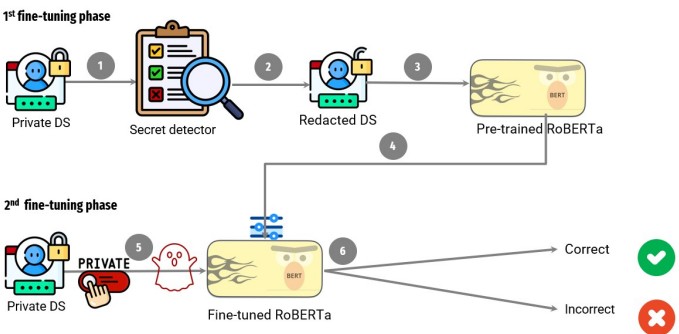

Figure 1: DualTune-GhostDP workflow.

To bridge this gap, we propose DualTune-GhostDP, a two-phase fine-tuning strategy that integrates ghost clipping for efficient gradient handling and EW-Tune for advanced privacy accounting, thereby mitigating privacy leakage while maintaining high utility.

## 4 Proposed Approach: DualTune-GhostDP

### 4.1 Motivation

Existing white-box approaches for privacy-preserving LLM fine-tuning improve specific aspects of the training process, including utility, memory efficiency, selective privacy, and privacy accounting (Duan et al. 2023; Li et al. 2021a; Shi et al. 2022; Hong et al. 2023; Behnia et al. 2022). However, these advances remain fragmented, leaving the privacy-utility trade-off far from optimal. Motivated by this gap, we propose the first unified approach that jointly integrates these complementary components within a two-phase fine-tuning pipeline, enabling systematic evaluation of their combined effect under DP. Our approach first performs privacy-free fine-tuning on sanitized data, followed by DP-SGD optimization with privacy accountant. The first phase serves as public privacy-reducing preprocessing (Parthasarathy et al. 2024) and consumes no privacy budget, while the second phase provides the formal $(\epsilon, \delta)$-differential privacy guarantee on the original dataset.

### 4.2 Model building blocks

The proposed solution **DualTune-GhostDP** integrates DualTune, a two-phase fine-tuning method, with GhostDP, which utilizes Ghost Clipping instead of the canonical clipping technique used in traditional DP. In addition, it utilizes the EW privacy accountant instead of the RDP that prior work used. The selection of this clipping technique and the privacy accountant is based on their demonstrated superiority over the other adopted methods, as shown in tables 6 and 7 in the Appendix, which compare the selected technique to the most commonly used methods in the literature. The DualTune-GhostDP approach consists of the following steps as shown in Figure 1:

1. Fine-Tuning on Redacted Data: The large pre-trained model is first fine-tuned on redacted (sanitized) data, generated by applying a high-contextual secret detector to the original dataset. This allows the model to learn useful task representations without the addition of privacy noise, ensuring that sensitive details are not memorized during training. The **Secret Detector** is a deterministic preprocessing function that removes predefined sensitive attributes, including named entities, pronouns, proper nouns, verbs, and syntactic roles (subjects/objects). It is implemented using SpaCy-based NER, POS tagging, and dependency parsing, and its output is treated as non-sensitive.

2. Fine-Tune using DP-SGD with Ghost Clipping: The original (unredacted) private data is used to further fine-tune the model obtained from Stage One. This is done under a DP setting, incorporating Ghost Clipping. The model is then re-evaluated to assess improvements.

3. Privacy controlling using EW Privacy Accountant: Optimize noise levels to add less noise than RDP while still ensuring DP guarantees. This improves model utility by accounting for finite training steps, rather than assuming unlimited iterations.

### 4.2.1 Ghost Clipping

Ghost Clipping Li et al. (2021a) is implemented in our approach using the `Private-Transformers` privacy engine with `clipping_mode="ghost"`. Ghost Clipping avoids materializing per-example gradients by computing clipping coefficients directly from activations and back-propagated signals, substantially reducing memory overhead. For a linear layer with activations $a_i$ and backpropagated error signals $\delta_i$ corresponding to training sample $i$, the per-example weight gradient is expressed as

$$g_i = \delta_i a_i^T. \tag{1}$$

Using properties of the Frobenius norm, the corresponding gradient norm can be computed without explicitly constructing $g_i$:

$$\|g_i\|_F^2 = \|\delta_i\|_2^2 \|a_i\|_2^2. \tag{2}$$

Ghost Clipping applies this computation across all trainable layers. The squared norm of the full per-example gradient is obtained by summing the layer-wise contributions,

$$\|g_i\|_2^2 = \sum_{l=1}^{L} \|g_i^{(l)}\|_2^2, \tag{3}$$

where $g_i^{(l)}$ denotes the gradient contribution of layer $l$. The clipping coefficient is then computed as

$$R_i = \min\left(1, \frac{C}{\|g_i\|_2}\right), \tag{4}$$

and the clipped gradient becomes

$$\tilde{g}_i = R_i g_i, \tag{5}$$

where $C = 0.1$ is the clipping bound used throughout all experiments. The clipped gradients are subsequently aggregated and perturbed using the Gaussian mechanism before updating the model parameters. This substantially reduces GPU memory consumption while preserving the same differential privacy guarantees as standard per-sample clipping.

### 4.2.2 Edgeworth Privacy Accountant

Privacy loss during the second fine-tuning phase is tracked using the Edgeworth (EW) privacy accountant which follows the procedure proposed by (Behnia et al. 2022). Unlike RDP, which relies on asymptotic moment-based approximations, the EW accountant employs Edgeworth expansions to approximate the privacy-loss distribution under finite compositions of subsampled Gaussian mechanisms Wang et al. 2019. During training, privacy loss is accumulated after every optimization step and converted into an $((\epsilon, \delta))$-DP guarantee. All experiments target a privacy budget of $\epsilon = 3$ and $\delta = 10^{-5}$. The accountant continuously monitors cumulative privacy loss and reports the final privacy guarantee at the end of training.

The motivation for using EW accounting is its ability to provide tighter finite-sample privacy estimates than RDP, allowing a more accurate characterization of privacy loss under practical LLM fine-tuning settings.

### 4.3 Problem formulation & Algorithmic Description

Let $\mathcal{D} = \{(x_i, y_i)\}_{i=1}^{N}$ denote a labeled dataset and $M_\theta$ a pretrained language model with parameters $\theta$. The objective is to minimize

$$\min_\theta \frac{1}{N} \sum_{i=1}^{N} \mathcal{L}(f_\theta(x_i), y_i)$$

while satisfying $(\epsilon, \delta)$-differential privacy with respect to the training dataset $\mathcal{D}$. Let $M_0$ denote a pretrained language model and $D$ the original training dataset. DualTune-GhostDP consists of three stages. **Sanitization:** A sanitized dataset $D_{\mathrm{san}} = f(D)$ is generated using a secret detector that removes predefined sensitive attributes. **Phase 1 (Public Fine-Tuning):** The model is fine-tuned on $D_{\mathrm{san}}$ using a standard optimizer, producing $M_1 = \mathrm{FineTune}(M_0, D_{\mathrm{san}})$. **Phase 2 (Private Fine-Tuning):** Starting from $M_1$, DP-SGD with Ghost Clipping is applied to the original dataset $D$. Per-example gradient norms are approximated, clipped to an $\ell_2$ bound $C$, aggregated, and perturbed with Gaussian noise calibrated by multiplier $\sigma$. **Privacy Accounting:** The cumulative privacy loss of Phase 2 is tracked using the Edgeworth (EW) accountant to obtain the final privacy budget $(\epsilon, \delta)$. The central insight of DualTune-GhostDP is that reducing gradient sensitivity prior to private optimization allows DP-SGD to operate in a lower-noise regime, improving model utility without weakening the formal differential privacy guarantees. Training follows a centralized pipeline controlled by a trusted server, enabling the model to learn general representations from sanitized data before privacy-preserving optimization. The resulting model $M^*$ satisfies differential privacy guarantees with respect to $D$, as summarized in Pseudocode 1.

---

**Pseudocode 1** Privacy-Preserving Dual-Phase Fine-Tuning (DualTune-GhostDP)

---

**Require:** Pre-trained RoBERTa model $M$, Dataset $D$
**Ensure:** Final fine-tuned model $M^*$
 1: Initialize $M \leftarrow$ Load RoBERTa
 2: $D \leftarrow$ Load and preprocess($D$)
 3: $D_{balanced} \leftarrow$ Balance($D$)
 4: $D_{sanitized} \leftarrow$ ApplySecretDetector($D_{balanced}$)
 5: $M \leftarrow$ FineTune($M, D_{sanitized}$)         ▷ First Fine-tuning Phase
 6: $M^* \leftarrow$ FineTuneDP($M, D_{balanced}$, ghost clipping)         ▷ Second Fine-tuning Phase
 7: Monitor $\epsilon \leftarrow$ EW($M^*$)
 8: Measure performance metrics: accuracy, execution time, memory usage
 9: Compare model accuracy under varying $\epsilon$ values

---

### 4.4 Privacy Analysis

**Differential Privacy Guarantee.** Phase 2 of DualTune-GhostDP is trained using DP-SGD with Ghost Clipping, clipping norm $C$, noise multiplier $\sigma$, and sampling rate $q$. When privacy loss is tracked using the Edgeworth (EW) accountant, the resulting model $M^*$ satisfies $(\epsilon, \delta)$-differential privacy with respect to the original dataset $D$.

**Loss function.** The model is trained using the standard cross-entropy loss

$$\mathcal{L}(x, y) = -\sum_{c=1}^{C} y_c \log p_\theta(c|x)$$

where $p_\theta(c|x)$ is the predicted probability of class $c$. During the DP phase, gradients of this loss are computed per example, clipped to an $\ell_2$ bound $C$, and perturbed with Gaussian noise before aggregation.

Ghost Clipping preserves the sensitivity of standard per-sample clipping by providing exact or conservative estimates of per-sample gradient norms (Lee & Kifer 2020), ensuring that each update corresponds to a Gaussian mechanism with bounded sensitivity. The EW accountant composes these mechanisms over a finite number of training steps using Edgeworth expansions, yielding tighter privacy estimates than RDP-based accounting. Since Phase 1 operates solely on sanitized data, it incurs no privacy cost and is implemented as a deterministic, rule-based and high contextual transformation that replaces sensitive tokens with abstract placeholders. This transformation reduces direct exposure of sensitive attributes but does not provide formal differential privacy guarantees, as implicit or contextual information may remain. Therefore, Phase 1 is best viewed as a privacy-reducing preprocessing; thus, the overall guarantee is fully determined by Phase 2.

### 4.5 Threat Model

The attacker is assumed to have black-box access to the model, meaning they can query the model and observe output predictions or confidence scores but do not have access to internal parameters, gradients, or training data. He may possess partial knowledge of the data distribution and may construct shadow models to simulate the training process.

Our defense mechanism is based on DP, which provides formal guarantees limiting the influence of any single training example on the learned model parameters. Under DP, the success probability of membership inference is provably bounded. Although our primary evaluation focuses on MIA, our approach also reduces information leakage that could enable reconstruction-based attacks.

To clarify communication security assumptions, our threat model focuses on privacy leakage from the trained model parameters and gradients during the training process. We assume a standard trusted training environment in which the optimization pipeline is executed locally or within a secure infrastructure. Attacks targeting the communication channel (e.g., man-in-the-middle attacks during distributed training) are orthogonal to the scope of this work and can be mitigated using established secure communication protocols such as TLS or secure aggregation. Our contribution instead addresses privacy leakage arising from model training dynamics and parameter updates.

## 5 Experiments and Discussion

In this section, we present four experiments designed to evaluate the effectiveness of each building block of our proposed method and to demonstrate improvements over prior work. All experiments are documented in the GitHub link provided at the end. Experiment 1 investigates the optimal epoch count for each dataset and analyzes how accuracy varies with this parameter for a single-phase DP-SGD model. Experiment 2 reproduces the approach of Shi et al. (2022), but substitutes the RDP accountant with the EW accountant to assess its impact. Experiment 3 evaluates our proposed method, DualTune-GhostDP, which builds on Experiment 2 by replacing standard clipping with Ghost Clipping. This modification emphasizes improvements in runtime, memory efficiency, convergence, and accuracy. Finally, Experiment 4 demonstrates the consistency of the privacy–utility trade-off on range of epsilon values achieved by our proposed method. It shows the consistent performance of DualTune-GhostDP under different epsilon values. The implementation and evaluation of **DualTune-GhostDP** are performed using an HPC cluster to ensure efficient code execution and result visualization.

### 5.1 Model, Datasets and Mechanism Selection

The **RoBERTa-base** model is chosen for its strong, well-established performance and its prevalent use in related literature, ensuring a fair basis for comparison. Additionally, it is well-suited for deployment in resource-constrained environments. We initialize the model using the publicly available RoBERTa-base checkpoint provided through the Hugging Face Transformers library. This model was originally pretrained on large-scale corpora including BookCorpus and Wikipedia using masked language modeling objectives. Three binary classification datasets from the GLUE benchmark were used for evaluating our suggested model: SST-2 (Stanford Sentiment Treebank 2), QNLI (Question Natural Language Inference) and QQP (Quora Question Pairs). **SST-2** is a sentiment classification dataset that contains movie reviews from Rotten Tomatoes, labeled as either positive (1) or negative (0), and is widely regarded for its effectiveness in evaluating sentiment analysis models. **QNLI** is a dataset containing questions/answers task classifying pairs as *Entailment* where the answer sentence logically answers the question or *Not Entailment* where it does not. **QQP** is a paraphrase detection task that detects *Duplicate* versus *Not Duplicate* question pairs. These datasets were chosen to assess the generalizability of our method across different natural language understanding tasks. Table 8 in the Appendix demonstrates some statistics on the used datasets.

All experiments in this work employ the *Gaussian differential privacy mechanism* implemented through *DP-SGD*, where Gaussian noise is added to the aggregated, clipped gradients after each mini-batch update. The privacy guarantees follow $(\epsilon = 3, \delta = 10^{-5})$-differential privacy, with a fixed noise multiplier of $\sigma = 1$ for Experiments 1–3 to enable consistent and fair comparison with prior work. Experiment 4 further explores different values of $\epsilon$ while maintaining the same $\delta$ and $\sigma$ to analyze the privacy–utility trade-off.

## 5.2 Experiments

To demonstrate the effectiveness of each building block, four experimental setups are developed.

To generate the redacted dataset used in experiments 2 (Two-phase fine-tuning with DP-SGD and standard clipping) and 3 (Two-phase fine-tuning with DP-SGD and Ghost clipping), SpaCy's off-the-shelf NLP tools were utilized, including Named Entity Recognition (NER), Part-of-Speech (POS) tagging, and dependency parsing. Since sentiment classification tasks often involve sensitive information that is semantically or contextually implied rather than explicitly stated, a High Contextual Secret Detector was adapted to redact a broader set of features, including: All 18 NER entity types, Pronouns (POS tag: PRON), Proper nouns (PROPN), All verbs (VERB), Subjects and objects (dependency tags: nsubj, dobj, pobj).

**Experiment 1: Baseline: Single-phase private fine-tuning using DP-SGD**
This experiment evaluates single-phase private fine-tuning using DP-SGD with standard clipping on SST-2, QNLI, and QQP. The number of training epochs is varied to identify optimal convergence under a fixed privacy budget ($\epsilon = 3, \delta = 10^{-5}$). Figure 4 in Appendix A.4.1 reports accuracy as a function of epoch count. The optimal epoch counts were four for SST-2 and QNLI, and two for QQP. These results establish the baseline performance used for comparison in subsequent experiments.

**Experiment 2: Two-phase fine-tuning with DP-SGD, standard clipping and EW Accountant**

In this experiment, the model undergoes two sequential fine-tuning phases. The first phase uses a public optimizer on redacted data where sensitive information is masked using a high-contextual detector without any privacy protection. The second phase applies DP-SGD with standard clipping on the original, unredacted data along with EW privacy accountant.

Each task (SST-2, QNLI, and QQP) is fine-tuned using varying combinations of epoch counts across the two phases, aiming to identify the optimal configuration that achieves the highest accuracy under a fixed privacy budget of $\epsilon = 3$. Epoch combinations start from (1,1), representing 1 epoch for each phase and increase in multiple of 2 until the maximum epoch count used in Experiment 1 (e.g., up to (4,4) for SST-2).

**Discussion:**
The optimal utility is achieved with fewer total epochs used in experiment 1 particularly at the (2,2) for SST-2, (4,4) for QNLI and at (2,1) for QQP configuration as shown in Tables 9, 10 and 11 which also corresponds to shorter training time. Beyond this point, additional epochs result in diminishing returns, where more computation time yields lower accuracy, making the setup both inefficient and suboptimal.
Comparing with experiment 1, the results indicate that the two-phase fine-tuning approach significantly outperforms the traditional single-phase fine-tuning with standard DP-SGD clipping under identical settings. Notably, this improvement in accuracy is achieved with fewer total epochs. For SST-2, accuracy increased by 3%, from 87.76.90% to 91.38%; for QNLI, by 5%, from 79.82% to 84.53%; and for QQP, by 6%, from 77.44% to 82.69% as shown from the tables.

A sanity check confirms that any configuration from Exp2 (not necessarily the optimal) achieves higher accuracy than all configurations in Exp1 across all datasets. These results highlight the effectiveness of introducing an initial redaction phase. This phase not only preserves privacy by removing sensitive information but also appears to enhance model performance and sufficiently initializes useful representations which advocates the idea of Tramer & Boneh (2020) that access to features learned on public data from a same domain can enhance privacy-preserving learning performance. The findings support the claim that sensitive data is not essential for effective training and that its careful removal can, in fact, be beneficial.
**Gradient Sensitivity Analysis:** Beyond accuracy improvements, we analyze gradient sensitivity during the private fine-tuning phase of Exp1 and Exp2 to better understand the effect of sanitized initialization on DP-SGD optimization. We summarize gradient statistics using the mean, median, and 95th percentile ($p95$) of per-example $\ell_2$ gradient norms, together with the clipping rate. These metrics jointly capture the central tendency, tail behavior, and the extent of clipping-induced distortion introduced by DP-SGD.

As shown in Appendix A.4.2 (Table 13), per-example gradient sensitivity statistics are reported for all evaluated datasets. To discuss SST-2 as a representative case, Exp2 substantially reduces per-example gradient sensitivity relative to Exp1: the mean gradient norm decreases by nearly an order of magnitude,

Table 1: Peak Memory Usage (MB): Normal vs. Ghost Clipping

| Dataset | Normal Clipping | Ghost Clipping | Reduction |
|---|---|---|---|
| **SST-2** | 24,880.69 | 8,694.16 | ∼3× |
| **QQP & QNLI** | 23,850.41 | 8,694.16 | ∼2.7× |

indicating a more stable optimization regime, while the median norm approaches zero, reflecting tightly concentrated per-example updates. In addition, the $p95$ norm is significantly reduced, evidencing a marked attenuation of heavy-tailed gradient behavior known to exacerbate DP-SGD clipping and noise amplification. This effect is further reflected in the clipping rate: whereas more than 41% of gradients are clipped in Exp1, only 15.82% are clipped in Exp2. Together, these results indicate that Exp2 preserves a larger fraction of the original gradient signal, thereby reducing clipping-induced bias and improving optimization efficiency under the same DP configuration.

**Experiment 3: Two-phase fine-tuning with DP-SGD and Ghost Clipping**

In this experiment, the setup and hyperparameters are identical to those used in Experiment 2, with the only difference being the clipping technique. Ghost Clipping is employed here instead of standard clipping. As before, each task (SST-2, QNLI, and QQP) is fine-tuned using various combinations of epoch counts across the two training phases, with the objective of identifying the optimal configuration that maximizes accuracy under a fixed privacy budget of $\epsilon = 3$.

Tables below summarize the best-performing epoch combinations for each DS. Highest accuracy was achieved using (2,1) epochs for SST-2 with 92.29%, (2,4) epochs for QNLI with 86.03% and (2,2) for QQP with 83.72%.

**Discussion:**

Compared to Experiment 2, Ghost Clipping provided significant memory efficiency gains, reducing usage by approximately three times from 24,880.69 MB for SST-2 and 23,850.41 MB for QNLI and QQP (under normal clipping) to a uniform 8,694.16 MB across all datasets as shown in Table 1. This memory reduction stems from the reduced need for large accumulated gradient buffers when using ghost norm estimation, which also accelerated model convergence. The identical memory usage observed under Ghost Clipping highlights its independence from dataset structure and sequence length, leading to equalized memory usage. In contrast, the higher memory consumption of SST-2 under normal clipping can be explained by its longer average input sequences (full movie review sentences), which inflate per-sample gradient storage requirements despite the dataset's smaller size. Meanwhile, QNLI and QQP, both sentence-pair tasks with similar average tokenized lengths, naturally exhibit comparable memory usage.

Consequently, the model achieved faster convergence in terms of runtime. Ghost Clipping reduced the training time by 25.6% for SST-2, 17.9% for QNLI, and 5.4% for QQP, as shown in Table 2. While QQP required the same number of epochs, both SST-2 and QNLI benefited from fewer total epochs count, further contributing to computational efficiency.

Among the datasets, SST-2 exhibited the most pronounced reduction. This can be attributed to the interaction between dataset size and clipping strategy. While SST-2 is smaller in scale compared to QNLI and QQP, the per-sample gradient computation in traditional DP-SGD introduces significant overhead. Ghost Clipping eliminates this overhead, leading to greater efficiency gains. In contrast, for larger datasets such as QQP, runtime is dominated by dataset scale that is the high number of training instances rather than per-sample operations, resulting in smaller relative improvements.

Table 2: Runtime Comparison: Normal vs Ghost Clipping

| Dataset | Normal Clipping | Ghost Clipping | Reduction |
|---|---|---|---|
| **SST-2** | 00:57:30 | 00:42:46 | ↓ 25.6% |
| **QNLI** | 03:28:20 | 02:51:06 | ↓ 17.9% |
| **QQP** | 05:19:29 | 05:02:05 | ↓ 5.4% |

Table 3: Performance comparison with prior work at $\epsilon = 3$. Best results are in **bold**.

| Model (Prior Work ↓, Dataset →) | SST-2 | QNLI | QQP |
|---|---|---|---|
| PromptDP-SGD | 90.48 | 83.62 | 80.29 |
| JFT | 89.22 | 84.02 | 84.77 |
| RoBERTa with DP-Adam | 86.12 | 84.62 | **85.41** |
| **Ours (EW Accountant + Ghost)** | **92.08±0.38** | **85.95±0.18** | 84.40±0.14 |

Table 4: Ablation study of DualTune-GhostDP on SST-2 ($\epsilon = 3$).

| Method | 2-Phase | Ghost | Acc. (%) | Time | Mem. (MB) |
|---|---|---|---|---|---|
| DP-SGD | × | × | 87.76 ± 1.43 | 00:51:33 | Not comparable |
| Two-Phase DP-SGD | ✓ | × | **91.38 ± 0.43** | 00:57:30 | 24880.69 |
| DualTune-GhostDP | ✓ | ✓ | **92.08 ± 0.38** | **00:42:46** | **8694.16** |

Interestingly, this reduction in memory was accompanied by improved optimal accuracy values across the three datasets as shown in Figure 3. In this experiment, Ghost Clipping was shown to enhance utility likely because it removes less private information and avoids unnecessary gradient suppression during updates, requiring less noise calibration.

**Experiment 4: Privacy-utility trade off**
The objective of this experiment is to evaluate the performance of our proposed method in balancing privacy and utility across different privacy budgets, ranging from 1 to 8. Since lower values of $\epsilon$ correspond to stronger privacy guarantees, often at the cost of reduced model accuracy, this study examines the robustness of our approach across this spectrum. We conduct the evaluation on two benchmark datasets SST-2 and QQP chosen to represent two datasets with different task characteristics and performance ranges.

**Discussion:**
Figure 2 demonstrates the privacy–utility behavior of our approach across different $\epsilon$. As expected, model accuracy generally improves as $\epsilon$ increases because less noise is injected during DP optimization.

For SST-2, the model maintains strong performance even under strict privacy constraints ($\epsilon = 1$), achieving 91.53% accuracy with minimal variation between seeds, which indicates robustness to privacy noise. As the privacy budget increases, the accuracy improves steadily and reaches 92.23% at $\epsilon = 8$, showing that the model benefits from relaxed privacy constraints while maintaining stable performance.

A similar trend is observed on QQP. The model achieves competitive accuracy even under strong privacy ($\epsilon = 1$) with 82.88%, and gradually improves to 84% at $\epsilon = 8$. This behavior is consistent with the expected privacy–utility trade-off where smaller $\epsilon$ provides stronger privacy protection but slightly reduces model utility, while larger $\epsilon$ improves predictive performance.

Importantly, across all evaluated privacy budgets, our approach consistently outperforms prior work while maintaining strong performance even at low $\epsilon$. The combination of the EW privacy accountant and Ghost Clipping within 2-phase fine-tuning strategy enables more efficient use of the privacy budget, resulting in a more favorable privacy–utility trade-off compared to standard DP-SGD approaches. Further analysis of the individual contributions of each component in the approach is provided in the ablation study (Table 4).

### 5.3 Ablation Study of DualTune-GhostDP

To quantify the contribution of each component in DualTune-GhostDP, we conduct an ablation study isolating the effects of the two-phase training pipeline and Ghost Clipping. Table 4 reports the results relative to a standard DP-SGD baseline. The results indicate that each component independently improves the privacy-utility trade-off, while their combination achieves the strongest performance and computational efficiency. Additional insight into the effect of the two-phase pipeline can be observed from the gradient sensitivity statistics reported in Appendix A.4.2 Table 13 where Two-Phase DP-SGD (Exp2) consistently reduces both the average gradient magnitude and the upper-tail distribution (p95) across datasets compared to the DP-

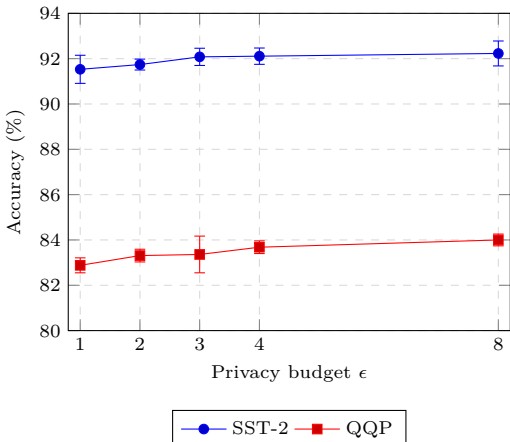

Figure 2: Privacy–utility trade-off for our model on SST-2 and QQP datasets across different privacy budgets $\epsilon$. Error bars indicate standard deviation over multiple seeds.

SGD baseline (Exp1). This behavior results in fewer clipping events under a fixed clipping norm, leading to more stable DP-SGD updates and improved downstream accuracy.

### 5.4 Comparison with Prior Work

We compare against three representative DP fine-tuning baselines covering single-phase, dual-phase, and memory-efficient optimization These findings strongly support the effectiveness of our suggested model that uses Ghost Clipping over standard clipping in the DualTune fine-tuning process and EW privacy accountant over RDP, offering advantages in memory usage, time, and utility. Table 3 shows a comparison between our suggested model and the three works in literature under same privacy budget and using the same pre-trained model RobERTa. **PromptDP-SGD:** (single-phase DP fine-tuning with RDP accountant) (Duan et al. 2023), **JFT** (Just fine tune twice,two-phase fine-tuning with RDP and normal Clipping) (Hong et al. 2023) and **RoBERTa with DP-Adam** (single phase fine-tuning using RDP and ghost clipping) (Li et al. 2021a). Our model, DualTune-GhostDP, clearly outperforms all the other works emphasizing the architecture benefits for the two datasets SST-2 and QNLI. However, for QQP, it outperforms PromptDP-SGD by 4%, while remaining comparable to the other two techniques, which could potentially surpass it when multiple runs and additional resources are considered.

Our results confirm the effectiveness of the proposed unified DualTune-GhostDP approach. The two-phase fine-tuning strategy combined with the EW Accountant, along with the integration of Ghost Clipping, yields a high-performing private model. This model outperforms both the two-phase variant with normal clipping under the RDP Privacy Accountant and the baseline single-phase private fine-tuning. The accuracy gains and their standard deviations over five runs, contributed by each component of our framework, are presented in Figure 3 and Table 5. Specifically, Experiment 2 extends the work of Shi et al. (2022) by replacing the privacy accountant, while Experiment 3 represents our complete proposed method, incorporating both the new privacy accountant and Ghost Clipping. Together, these experiments validate our DualTune-GhostDP framework as a high-performing, fast, efficient, and strong privacy-preserving fine-tuning unified strategy. It achieves strong utility without compromising privacy, efficiency, or scalability.

While RoBERTa is used for consistency with prior work, DualTune-GhostDP is model-agnostic and applies to any transformer fine-tuned with DP-SGD. The reliance on RoBERTa reflects experimental control rather than architectural dependence. All components operate independently of model architecture.

### 5.5 Generalization to Other Transformer-based Backbones

To assess whether the observed benefits of DualTune-GhostDP are specific to RoBERTa or extend to other transformer architectures, we repeat our experiments using BERT-base (uncased) under the same privacy

Table 5: Accuracy Evolution Across All Experiments (mean ± std).

| Dataset | E1:Single (DP-SGD) | E2:Dual+Normal+EW | E3:Dual + Ghost |
|---------|--------------------|--------------------|-----------------|
| SST-2 | 87.76 ± 1.43 | 91.38 ± 0.43 | 92.08 ± 0.38 |
| QNLI | 79.82 ± 0.34 | 84.80 ± 0.33 | 85.95 ± 0.18 |
| QQP | 77.44 ± 0.87 | 82.69 ± 0.21 | 84.40 ± 0.14 |

Figure 3: Accuracy evolution across all experiments (mean ± std).

budget and comparable training settings. Table 12 in the Appendix isolates the incremental contributions of each training stage. Transitioning from single-phase DP-SGD (E1) to dual-phase training (E2) yields a substantial accuracy improvement of 5.02%, while introducing ghost clipping (E3) preserves accuracy and reduces peak GPU memory usage over E2 by approximately 4 times which confirms that each stage contributes a distinct and complementary benefit.

### 5.6 Membership Inference Attack (MIA) Results

We evaluate the privacy guarantees of DualTune-GhostDP under a strong MIA setting, where an adversary attempts to determine whether a specific sample was used during training based on model outputs. Following standard practice for DP-SGD evaluation, we consider a direct MIA on the SST-2 dataset, in which the adversary has access to model posteriors and ground-truth membership labels, representing a worst-case threat model (Shokri et al. 2017; Carlini et al. 2022a). Results show that the post-sanitization model and the full DualTune-GhostDP pipeline both achieve attack performance indistinguishable from random guessing, while a single-phase DP-SGD baseline exhibits noticeably higher and less stable attack success. These findings confirm that combining data sanitization with DP-SGD and Ghost Clipping effectively suppresses membership leakage, yielding stronger privacy protection without sacrificing model utility. A detailed experimental setup is described in the Appendix A.5 and Table 14. While our evaluation focuses on MIA, other threats such as evasion attacks and data poisoning attacks may also affect ML systems. DP provides partial robustness against these threats by limiting the influence of individual samples during training.

## 6 Conclusion

We proposed DualTune-GhostDP, a unified framework combining two-phase fine-tuning, Ghost Clipping, and the Edgeworth accountant for differentially private LLM fine-tuning. Experiments demonstrate improved utility, convergence, runtime, and memory efficiency while preserving rigorous differential privacy guarantees.Experimental results demonstrate that DualTune-GhostDP maintains strong model performance while saving memory, reducing runtime, speeding up convergence and incorporating privacy compared to prior works. **Limitations and Future Work.** In our experiments, we fine-tuned the encoder-only models as the pre-trained backbone to maintain consistency with prior work, but exploring other modern decoder-only LLMs is our future direction which raises additional architectural considerations related to parameter sharing and Ghost Clipping efficiency, which we discuss in Appendix.

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

# A    Appendix: Supplementary Material

This appendix provides additional theoretical justification, dataset statistics, extended experimental results, and detailed privacy and attack evaluations omitted from the main paper due to space constraints. All results are reported to ensure reproducibility and completeness.

## A.1    Comparison of Privacy Accounting and Gradient Clipping Mechanisms

This section provides a comparative analysis of the core building blocks adopted in DualTune-GhostDP against traditional alternatives commonly used in differentially private training.

Table 6: Comparison between Rényi Privacy Accounting (RDP) and Edgeworth Privacy Accounting (EW).

| Aspect | Rényi DP (RDP) | Edgeworth (EW) |
|---|---|---|
| Privacy regime | Asymptotic (many iterations) | Finite-sample training |
| Composability | Moment-based composition | Edgeworth expansion |
| Noise calibration | Conservative | Tighter for same $\epsilon$ |
| Accuracy–privacy trade-off | Lower utility | Improved utility |
| Best suited for | Large-scale DP training | DP fine-tuning of LLMs |

Table 7: Comparison between Standard DP-SGD Clipping and Ghost Clipping.

| Aspect | Standard DP-SGD | Ghost Clipping |
|---|---|---|
| Gradient computation | Explicit per-sample gradients | Implicit norm estimation |
| Computational cost | High | Low |
| Memory usage | High | Low |
| Accuracy impact | Standard DP trade-off | Comparable with efficiency gains |

## A.2    Dataset Statistics

Table 8 reports detailed statistics for all datasets used in this work, including task type, dataset size, and balanced training splits.

Table 8: GLUE Dataset Statistics with Task Types and Balanced Train Sizes

| Dataset | Task Type | Split | Size (MB) | #Samples |
|---|---|---|---|---|
| SST-2 | Sentiment Analysis | Train | 11.82 | 67,349 (59,560 balanced) |
| | | Validation | 0.15 | 872 |
| | | Test | 0.32 | 1,821 |
| QQP | Paraphrase Detection | Train | 63.85 | 363,846 (268,756 balanced) |
| | | Validation | 7.09 | 40,430 |
| | | Test | 68.61 | 390,965 |
| QNLI | QA/NLI | Train | 18.38 | 104,743 (104,732 balanced) |
| | | Validation | 0.96 | 5,463 |
| | | Test | 0.96 | 5,463 |

Table 9: Accuracy Variation with Epoch Combinations for SST2

| # Epochs (Phase 2 ↓, Phase 1 →) | 1 | 2 | 4 |
|---|---|---|---|
| 1 | 90.42% | 89.72% | 91.00% |
| 2 | 91.36% | **91.38**% | 90.77% |
| 4 | 89.72% | 91.12% | 91.36% |

### A.3 Extended Accuracy Results and Epoch Sensitivity

### A.3.1 Accuracy Variation with Epoch Combinations

Figures 9, 10 and 11 detail the full accuracy vs epoch sensitivity.

Table 10: Accuracy Variation with Epoch Combinations for QNLI

| # Epochs (Phase 2 ↓, Phase 1 →) | 1 | 2 | 4 |
|---|---|---|---|
| 1 | 83.49% | 81.94% | 83.64% |
| 2 | 83.31% | 83.59% | 83.88% |
| 4 | 84.14% | 84.2% | **84.53**% |

Table 11: Accuracy Variation with Epoch Combinations for QQP

| # Epochs (Phase 2 ↓, Phase 1 →) | 1 | 2 |
|---|---|---|
| 1 | 82.53% | 73.39% |
| 2 | 82.16% | **82.69**% |

### A.3.2 Generalization to BERT-base (Uncased)

To assess the generality of the proposed framework beyond the primary backbone, we evaluate DualTune-GhostDP on **BERT-base (uncased)** under the same privacy budget with the results documented in table 12.

### A.4 Additional Experimental Results

### A.4.1 Experiment 1: Single-Phase DP-SGD Baseline

This experiment establishes the baseline against which all subsequent dual-phase results are compared.

In this experiment, the model undergoes a single phase of fine-tuning using DP-SGD with standard clipping using $(\epsilon, \delta)$ values as declared above on the original training datasets for the three tasks (SST-2, QNLI, and QQP). During this stage, the per-sample gradients are clipped to an $\ell_2$-norm bound of 0.1 to control sensitivity, after which Gaussian noise is added to ensure differential privacy guarantees. The clipping is applied at the gradient level rather than on the input space, following the conventional DP-SGD formulation. The number of training epochs is varied to determine the optimal epoch count that yields the highest accuracy for each dataset. This setup serves as the baseline for privacy-preserving fine-tuning approaches against which subsequent experiments are compared. The results, illustrated in Figure 4, show the relationship between

Table 12: Targeted improvements on **BERT-base (uncased)** for SST-2 at $\epsilon = 3$: accuracy gain from E1→E2 and memory reduction from E2→E3.

| Improvement Target | Transition (Before → After) | Net Change |
|---|---|---|
| Accuracy (%) | 85.51 → 90.53 | ↑ **+5.02** |
| Peak GPU Reserved (MB) | 27748.00 → 7038.00 | ↓ **−20710.00** |

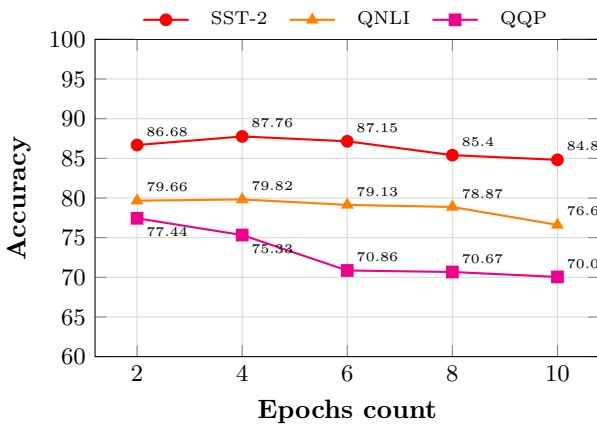

Figure 4: Accuracy variation vs epochs count for each DS

accuracy and the number of training epochs for the SST-2, QNLI, and QQP datasets. The optimal epoch counts observed were four for SST-2 with an accuracy of **87.76%**, four for QNLI with an accuracy of **79.82%**, and two for QQP with an accuracy of **77.44%**.

This pattern can be partly explained by the impact of dataset size on training under DP-SGD: larger datasets often lead to improved model utility within a fixed privacy budget, as each training step benefits from more diverse examples, potentially requiring fewer epochs to converge. However, excessive training can harm the learning process, leading to overfitting and a subsequent decrease in accuracy.

### A.4.2 Per-Example Gradient Sensitivity Statistics

To analyze how the proposed two-phase training strategy reshapes optimization under differential privacy, we report per-example gradient sensitivity statistics for all evaluated datasets (SST-2, QQP, and QNLI). These statistics are collected during the DP training phase and summarize the distribution of per-example gradient norms together with the fraction of gradients affected by clipping.

Table 13 reports four DP-relevant measures: the mean, median, and 95th percentile ($p95$) of per-example gradient norms, as well as the clipping rate. Collectively, these metrics capture (i) the overall scale of gradient sensitivity, (ii) the typical per-example contribution, (iii) the heaviness of the gradient tail, and (iv) the degree of optimization distortion introduced by DP clipping.

Across SST-2 and QQP, Exp2 substantially reduces the mean and $p95$ gradient norms while also lowering the clipping rate. This indicates a shift toward a lower-sensitivity, lighter-tailed gradient regime, in which fewer examples are aggressively rescaled by the clipping operation. Such behavior directly limits clipping-induced bias and is consistent with the observed gains in validation accuracy.

QNLI exhibits a distinct but still informative behavior under Exp2. While absolute gradient magnitudes are drastically reduced (mean 3.92, $p95$ 9.05), all gradients exceed the fixed clipping threshold ($C = 0.1$), resulting in a 100% clipping rate. This outcome reflects a scale mismatch between the dataset-specific gradient distribution and the globally fixed clipping bound, rather than instability or heavy-tailed behavior. Notably, the narrow gap between the median and $p95$ norms indicates a highly concentrated and well-controlled

Table 13: Per-example gradient sensitivity statistics during the DP phase across datasets. Exp2 consistently reduces gradient magnitudes and tail heaviness relative to Exp1, with dataset-dependent effects on clipping behavior under a fixed clipping bound.

| Dataset | Exp | Mean | Median | $p95$ | Clip (%) |
|---------|------|-------|--------|--------|----------|
| SST-2 | Exp1 | 302.9 | 1.72 | 1321.1 | 41.14 |
|       | Exp2 | 43.7 | 0.0036 | 191.3 | 15.82 |
| QQP | Exp1 | 310.2 | 2.06 | 1796.1 | 49.96 |
|     | Exp2 | 98.3 | 0.31 | 518.8 | 30.05 |
| QNLI | Exp1 | 298.8 | 3.75 | 1747.9 | 52.14 |
|      | Exp2 | 3.92 | 3.46 | 9.05 | 100.00 |

Table 14: Comparison of MIA Evaluation Metrics Across Model Pipelines on SST-2.

| Metric | Post-Sanitization | DualTune-GhostDP | Baseline Model |
|--------|-------------------|------------------|----------------|
| **Accuracy** | 0.512 | 0.508 | 0.501 |
| **AUC** | 0.5150 | 0.5033 | 0.5909 |
| **Precision** | 0.5071 | 0.5045 | 0.5082 |
| **Recall** | 0.8741 | 0.9189 | 0.1206 |
| **F1-Score** | 0.6418 | 0.6514 | 0.1950 |

gradient distribution, implying that gradients are uniformly rescaled rather than selectively distorted by clipping.

Overall, these results show that Exp2 consistently reshapes the per-example gradient landscape toward lower sensitivity and reduced tail risk across datasets. At the same time, they highlight how dataset-dependent gradient scales interact with fixed DP hyperparameters, emphasizing the importance of distributional diagnostics beyond clipping rate alone.

### A.5 Membership Inference Attack Evaluation

To provide a complete assessment of the privacy guarantees of DualTune-GhostDP, this section presents the full Membership Inference Attack (MIA) evaluation across different stages of the training pipeline. MIA is a privacy attack in which an adversary attempts to determine whether a given data sample was included in the model's training set based solely on the model's outputs. Models that memorize training data tend to exhibit higher MIA success rates, whereas privacy-preserving mechanisms, such as dataset sanitization and differential privacy, aim to reduce attack performance toward random guessing ($\approx 50\%$). In all experiments, we adopt a strong adversarial setting consistent with standard practice in DP-SGD privacy analysis. Specifically, we simulate a *direct membership inference attack* on the SST-2 dataset, which is widely considered one of the strongest threat models for evaluating privacy leakage in classification settings Shokri et al. (2017); Carlini et al. (2022a). In this setting, the adversary (i) has access to the model's output posteriors and (ii) knows the true membership status of samples used during training. This configuration provides an upper bound on the adversary's advantage and represents a worst-case estimate of potential privacy leakage.

#### A.5.1 MIA on Stage 1 Model: Post-Sanitization Model

The first evaluation is conducted on the model trained solely on the sanitized dataset, prior to applying any differentially private optimization. Under this setting, the attack achieves an accuracy of 0.512 and an AUC

of 0.5150, both of which are effectively equivalent to random guessing. Although the recall appears high, this behavior is attributed to the attacker over-predicting the 'member" class, which results in a large number of false positives as reflected in the confusion matrix. Importantly, the precision remains close to 0.50, indicating that the adversary cannot reliably distinguish between member and non-member samples. These results suggest that dataset sanitization alone already provides a strong degree of protection by reducing memorization and limiting privacy leakage. Consequently, the post-sanitization model serves as a robust initialization point for subsequent private fine-tuning.

### A.5.2  MIA on Proposed Approach: DualTune-GhostDP Model

We next evaluate the full DualTune-GhostDP pipeline, which combines dataset sanitization with DP-SGD using Ghost Clipping and the EW privacy accountant. Under the same direct-MIA setting, the resulting model achieves an attack accuracy of 0.508 and an AUC of 0.5033. These values correspond to pure random guessing, demonstrating that the adversary gains no meaningful advantage even under this strong threat model. The near-random attack performance confirms that integrating differential privacy on top of sanitized training further suppresses membership leakage. Compared to the post-sanitization model, the slight reduction in attack success indicates that DP-SGD with Ghost Clipping provides an additional layer of protection, reinforcing the privacy guaranties of the proposed framework.

### A.5.3  MIA on Baseline: Single-Phase DP-SGD Model

For comparison, we evaluate a baseline model trained using standard single-phase DP-SGD without dataset sanitization or Ghost Clipping. Despite operating under the same privacy budget, this baseline exhibits noticeably weaker and less stable privacy behavior. In one representative run, the attack AUC reaches 0.5909, indicating that the adversary can more reliably distinguish training samples compared to both the post-sanitization model and DualTune-GhostDP. Even in configurations where the baseline AUC approaches 0.50, the attack metrics remain highly unstable, with extremely low recall and F1 scores. This instability highlights the baseline model's reduced robustness to membership inference and suggests that standard DP-SGD alone may be insufficient to consistently suppress membership leakage in practice.

### A.5.4  Discussion

Across all evaluated settings, DualTune-GhostDP consistently provides the strongest protection against membership inference. Dataset sanitization substantially reduces memorization in the initial training stage, while the subsequent application of DP-SGD with Ghost Clipping further suppresses residual leakage. In contrast, the single-phase DP-SGD baseline demonstrates higher attack success and instability, despite operating under the same privacy budget. Overall, these results demonstrate that the combined use of sanitization and differentially private fine-tuning yields privacy guarantees that are indistinguishable from random guessing under a worst-case direct-MIA setting, while simultaneously maintaining higher model utility. This highlights the advantage of the proposed DualTune-GhostDP framework in achieving a more favorable privacy–utility trade-off than existing baselines.

### A.6  Future Work: Architectural Considerations for Decoder-Only LLMs

While our experiments focus on encoder-only backbones for consistency with prior DP fine-tuning work, extending DualTune-GhostDP to modern decoder-only LLMs (e.g., GPT-style architectures) introduces additional architectural considerations. In particular, decoder-only models rely heavily on parameter sharing mechanisms such as tied input and output embeddings and repeated reuse of projection matrices across layers which complicate the assumptions underlying Ghost Clipping's efficiency gains. Although Ghost Clipping remains theoretically applicable under DP-SGD, the presence of extensive parameter sharing can mask its practical memory and runtime benefits by collapsing multiple gradient paths onto shared parameters. We are actively investigating decoder-only variants that explicitly account for these interactions, including controlled untying and architectural adaptations, and leave a full empirical study of decoder-only LLMs to future work.

