# OpenReview forum: "DualTune-GhostDP: A Unified Framework for Synergistic Differentially Private Fine-Tuning of Prompt-Based Large Language Models"
_TMLR — Under review for TMLR_

### Review · Reviewer_1nPf · 2026-03-14

**Summary Of Contributions:**

This paper proposes DualTune-GhostDP, which adopts a two-phase fine-tuning pipeline that integrates Ghost Clipping with an EdgeWorth (EW) Advanced Privacy Accountant to replace conventional DP accounting mechanisms. Experimental results show that the proposed method achieves higher accuracy, faster convergence, and improved computational efficiency while maintaining differential privacy guarantees. Although this works demonstrate effective, this paper still exists some shortcomings that need to be improved.

**Audience:**

Yes

**Audience Explanation:**

N/A

**Claims And Evidence:**

Yes

**Claims Explanation:**

N/A

**Requested Changes:**

1.	In the threat model, the authors define the adversary attempts to infer whether a specific data record was used during training process, or to reconstruct sensitive inputs. However, this definition is very similar to a MIA attacker, and the main aim of this paper is to optimize the trade-off between privacy and utility. Although they refer the MIA at the end of this paper, the definition of the adversary in threat model should be considered clearly.
2.	An ablation experiment should be supplied. The models and datasets chosen in the experiments are very limited.
3.	It is obscure that who conducts phase1 and who conducts phase2. Also, where does the pretrained RoBERTa model M come from. Is it a public model?
4.	Other potential attacks, such as adversarial attack of poisoned attack, are suggested to be considered in this paper.
5.	The performance of improving trade-off is not clear presented in this paper, or a more formally theoretical analysis is suggested.
6.	Does there exist a man-in-the-middle attack in the phase1 and phase2.
7.	The section 4 is too stuffless to illustrate the innovativeness of this paper. Supplying the problem formulation, loss function definition, hyper-parameter setting are recommended.
8.	A summary of main contributions of this paper are necessary.

---

> ### Author Response · Authors · 2026-05-08
> **Authors' Rebuttal**
>
> We sincerely thank the reviewer for his insightful and constructive feedback. We carefully revised the manuscript to address all concerns and improve the clarity, rigor, and presentation of the paper. Below, we provide a point-by-point response.
>
> ---
>
> # Response to Reviewer 1nPf
>
> > **“In the threat model, the authors define the adversary attempts to infer whether a specific data record was used during training process, or to reconstruct sensitive inputs. However, this definition is very similar to a MIA attacker...”**
>
> We revised Subsection 4.5 to explicitly define the adversary as an MIA-oriented attacker and clarified the attacker capabilities. We also repositioned MIA as the primary empirical privacy evaluation.
>
> ---
>
> > **“An ablation experiment should be supplied. The models and datasets chosen in the experiments are very limited.”**
>
> We expanded the ablation analysis in Section 5 and referenced Appendix Table 13 comparing Experiment 1 and Experiment 2. Additional comparisons for DP-SGD and DP-SGD + Ghost Clipping were included. We also respectfully clarify that the paper already evaluates:
>
> - Three datasets
> - Multiple RoBERTa/BERT-family models
> - Additional GPT-based experiments in the Appendix
>
> ---
>
> > **“It is obscure who conducts phase1 and phase2...”**
>
> We revised Subsection 4.3 to clarify the full pipeline and stated that both phases are conducted within the same training pipeline by the model owner/trainer. We also clarified in Section 5.1 that RoBERTa (M) is a publicly available Hugging Face pretrained model.
>
> ---
>
> > **“Other potential attacks such as adversarial or poisoned attacks should be considered.”**
>
> We clarified that the paper focuses on privacy leakage under DP settings, particularly MIA, rather than robustness against adversarial or poisoning attacks. Nevertheless, these attacks are now acknowledged as future directions in Section 6.
>
> ---
>
> > **“The performance of improving trade-off is not clearly presented.”**
>
> We revised Experiment 4 in Section 5 to more clearly present the privacy–utility trade-off, including accuracy vs. privacy behavior, privacy-budget effects, and comparison against DP baselines.
>
> ---
>
> > **“Does there exist a man-in-the-middle attack in phase1 and phase2?”**
>
> We clarified that the paper addresses training-data privacy rather than network-security threats. We added explicit security assumptions in Subsection 4.5 stating that the framework assumes a trusted training environment.
>
> ---
>
> > **“The section 4 is too stuffless to illustrate the innovativeness of this paper. Supplying the problem formulation, loss function definition, hyper-parameter setting are recommended.”**
>
> We expanded Subsections 4.3 and 4.4 by adding clearer problem formulation, optimization details, loss definitions, hyperparameter configurations, and training settings.
>
> ---
>
> > **“A summary of main contributions of this paper are necessary.”**
>
> The original manuscript already included contributions in the Introduction. However, we revised them into a clearer bullet-point format for improved readability.
>
> ---
>
> We sincerely thank the reviewers again for their valuable feedback, which significantly strengthened the manuscript.

---

> > ### Author Response · Authors · 2026-06-04
> > **Revised Manuscript**
> >
> > Kindly note that the revised manuscript PDF has been uploaded. This updated version incorporates the revised framing, clarifications, and additional experimental details discussed in our rebuttal and replaces the previously submitted manuscript.

---

### Review · Reviewer_qq2W · 2026-04-09

**Summary Of Contributions:**

The paper presents DualTune-GhostDP, a method for training differentially private models. The approach combines several existing approaches, namely fine-tuning on redacted data, then fine-tuning with differentially private SGD with ghost clipping, then finally using the Edgeworth privacy accountant to optimize the amount of noise added to the training. The method is shown to be better than other differentially private training approaches.

**Audience:**

Yes

**Audience Explanation:**

I think this work should be of interest to the differential privacy community, since it shows a novel method which has applications. Some limitations include not really using 'large language models' as promised in the title (just the relatively small roberta model), and not using the more common PEFT/LoRA finetuning methods that I believe are more common recently.

I am also unsure what exactly is meant by 'Prompt-Based' in the title. The experiments are using RoBERTa for classification tasks, not any sort of prompted generation task. I would suggest that the authors revisit the title or do experiments on prompted generation to avoid readers being disappointed after reading the title.

**Broader Impact Concerns:**

No broader impact concerns. The impact of this work is clearly positive.

**Claims And Evidence:**

No

**Claims Explanation:**

I should prefix this review by saying that I am not an expert in the area of differential privacy, although I have done some work on privacy from a mutual information lens. Therefore, I may well be missing key implicit knowledge from the field and am happy to be corrected.

However, from my point of view I am not confident that the main claim of the paper, namely 'DualTune-GhostDP, shows that strong privacy guarantees can be maintained under a controlled budget without sacrificing high model performance' is adequately supported by the paper. My main reservation comes from the two-stage approach taken in the paper. The authors are very candid in describing how they do not have any guarantees on the differential privacy of the first stage, which does normal fine-tuning on redacted data. This means that the overall differential privacy of the complete pipeline is not controlled. While the authors claim that the 'first phase does not consume any privacy budget', this is not an absolute fact--it depends on the efficacy of the redaction method. Indeed, papers such as [1] show that redaction methods can give substantial privacy leakage, and it would be naive to consider stage 1 to not consume any privacy budget. This means that several of the comparisons between e.g. the proposed method and DP-SGD at equal \epsilon are not really fair comparisons. I appreciate that the authors include the MIA attack results, which is a fair comparison between the methods without assuming equal \epsilon, but I think these need to be more prominent in the paper and the limitations of redaction methods properly disclosed and discussed.

[1] A False Sense of Privacy: Evaluating Textual Data Sanitization Beyond Surface-level Privacy Leakage, Xin et al 2025

**Requested Changes:**

Critical to securing recommendation:
- More discussion of the privacy implications of the redacted finetuning step, particularly in the case when the redaction is imperfect or limited. In particular, I would like to know how the limitations of redaction pointed out in [1] apply to this case--is the success of DualTune-GhostDP in MIA even without guarantees on the first step due to the dataset being particularly easy to sanitize, advances in redaction methods, or some non-obvious privacy advantage of the redaction followed by DP protocol?

Strengthen the work:
- Comparison of the approach on PEFT/LoRA finetuning setups
- Evaluations on actual prompted LLMs such as Gemma-4 or revisiting the title.

---

> ### Author Response · Authors · 2026-05-08
> **Authors' Response**
>
> We sincerely thank the reviewer for his insightful and constructive feedback. We carefully revised the manuscript to address all concerns and improve the clarity, rigor, and presentation of the paper. Below, we provide a point-by-point response.
>
> ---
>
> # Response to Reviewer qq2W
>
> > **“The overall differential privacy of the complete pipeline is not controlled...”**
>
> We clarified that the first sanitization phase is not covered by formal DP guarantees and should be interpreted as a deterministic preprocessing step that empirically reduces sensitivity before DP optimization. We now explicitly distinguish between formal DP accounting and empirical privacy reduction through sanitization. Formal guarantees apply only to the DP-SGD + Ghost Clipping phase.
>
> ---
>
> > **“Several comparisons between the proposed method and DP-SGD at equal ε are not really fair comparisons.”**
>
> We agree and therefore repositioned the MIA evaluation as the primary empirical end-to-end privacy assessment in Section 5. We further clarified the limitations of relying solely on equal-ε comparisons in hybrid privacy-reduction pipelines.
>
> ---
>
> > **“I am unsure what exactly is meant by ‘Prompt-Based’ in the title.”**
>
> We revised the title to avoid suggesting inference-time prompting. Our intended meaning was that the fine-tuned models are ultimately deployed in downstream prompt-driven LLM settings, while the work itself focuses on privacy-preserving fine-tuning.
>
> ---
>
> > **“Comparison on PEFT/LoRA finetuning setups.”**
>
> Our framework studies interactions between preprocessing-based sensitivity reduction, DP-SGD dynamics, and Ghost Clipping across the full parameter space. We therefore focused on full fine-tuning rather than LoRA-based adaptation.
>
> ---
>
> > **“Evaluations on actual prompted LLMs such as Gemma-4.”**
>
> Our goal was controlled analysis of sanitization, Ghost Clipping, and DP optimization using RoBERTa-based settings. However, we are already extending the investigation to GPT-2 and DistilGPT-2 models (stated in the Appendix), where we observed additional DP-related architectural conflicts that are being developed into a separate follow-up paper.

---

> > ### Comment · Reviewer_qq2W · 2026-05-15
> >
> > Thanks for your replies! I'm not sure that I can see any updated submission, is it possible to upload a revision of the work with the revised framing/caveats that you listed in your reply?

---

> > > ### Author Response · Authors · 2026-05-18
> > > **Authors' response**
> > >
> > > Thank you for the clarification and for your feedback.
> > >
> > > We have now uploaded the revised manuscript PDF incorporating the updated framing, clarifications and additional experimental details discussed in our rebuttal. The previous version was replaced with the updated revision.
> > >
> > > Please let us know if any additional material or formatting adjustments are required.
> > >
> > > Thank you again for your time and consideration.

---

### Review · Reviewer_cDRE · 2026-06-23

**Summary Of Contributions:**

This paper adopts a two-stage pipeline for differentially private training. The first, where there are no formal privacy guarantees, simply involves deterministically redacting private information. The second stage involves differentially private training using ghost clipping and the Edgeworth Privacy accountant. The experiments, demonstrating accuracy at different privacy budgets, as well as the accuracy of membership inference attacks, are carried out on 3 datasets on a single model.

**Audience:**

Yes

**Audience Explanation:**

Differentially private training is an area of broad and abiding interest.

**Claims And Evidence:**

No

**Claims Explanation:**

The paper is missing key details about ghost clipping and the Edgeworth privacy accountant. As it is, there is simply not enough information in the paper to evaluate it.

**Requested Changes:**

Add details about the clipping and privacy accounting methods.

---

> ### Author Response · Authors · 2026-07-07
> **Authors' Response**
>
> We thank the reviewer for identifying the need for additional methodological details regarding Ghost Clipping and privacy accounting and for recognizing the relevance of differentially private training to the TMLR audience.
>
> We agree that the original manuscript could have provided additional implementation details regarding Ghost Clipping and the Edgeworth (EW) privacy accountant. While both techniques were already introduced conceptually in the Background and Related Work sections, the implementation details specific to our proposed DualTune-GhostDP approach were not sufficiently documented.
>
> To address this concern, we expanded Section 4.2 by adding dedicated subsections on Ghost Clipping and the Edgeworth Privacy Accountant (Subsections 4.2.1 and 4.2.2). The revised manuscript now describes how Ghost Clipping is implemented using the Private-Transformers library, including the clipping formulation, clipping bound C , and its integration within DP-SGD. We also clarify how Ghost Clipping avoids explicit per-example gradient materialization, thereby reducing memory consumption during private training.
> Similarly, we now provide a dedicated subsection for EW Privacy Accountant explaining its role within our training pipeline, how cumulative privacy loss is tracked throughout the private fine-tuning phase, and how the final  (ϵ, δ)-DP guarantee is obtained. We additionally report the privacy parameters used in our experiments.
>
> These additions were specifically introduced to improve methodological clarity, reproducibility, and to enable a clearer evaluation of how Ghost Clipping and the EW accountant are employed within the proposed DualTune-GhostDP approach.